



# Drought in a human-modified world: reframing drought definitions, understanding and analysis approaches

**Anne F. Van Loon[1],** Tom Gleeson[2], Julian Clark[3], Albert I.J.M. Van Dijk[4], Kerstin Stahl[5], Jamie Hannaford[6], Giuliano Di Baldassarre[7], Adriaan J. Teuling[8], Lena M. Tallaksen[9], Remko Uijlenhoet[8], David M. Hannah[1], Justin Sheffield[10], Mark Svoboda[11], Boud Verbeiren[12], Thorsten Wagener[13,14], Sally Rangecroft[1], Niko Wanders[10], Henny A.J. Van Lanen[8]

[1] Water Science Research Group, School of Geography, Earth, and Environmental Sciences, University of Birmingham, Edgbaston, Birmingham, B15 2TT, UK.
[2] Department of Civil Engineering, University of Victoria, Canada.
[3] Human Geography Research Group, School of Geography, Earth, and Environmental Sciences, University of Birmingham, UK.
[4] Fenner School of Environment & Society, the Australian National University, Canberra, Australia.
[5] Hydrology, Faculty of Environment and Natural Resources, University of Freiburg, Germany.
[6] Centre for Ecology and Hydrology, Wallingford, UK.
[7] Department of Earth Sciences, Uppsala University, Sweden.
[8] Hydrology and Quantitative Water Management group, Wageningen University, the Netherlands.
[9] Department of Geosciences, University of Oslo, Norway.
[10] Civil and Environmental Engineering, Princeton University, USA.
[11] National Drought Mitigation Center, University of Nebraska, Lincoln, USA.
[12] Department of Hydrology and Hydraulic Engineering, Vrije Universiteit Brussel, Belgium.
[13] Department of Civil Engineering, University of Bristol, UK.
[14] Cabot Institute, University of Bristol, UK.

*Correspondence to*: Anne F. Van Loon (a.f.vanloon@bham.ac.uk)

**Abstract.** In the current human-modified world, or 'Anthropocene', the state of water stores and fluxes has become dependent on human as well as natural processes. Water deficits (or droughts) are the result of a complex interaction between meteorological anomalies, land surface processes, and human inflows, outflows and storage changes. Our current inability to adequately analyse and manage drought in many places points to gaps in our understanding and to inadequate data and tools. The Anthropocene requires a new framework for drought definitions and research. Drought definitions need to be revisited to explicitly include human processes driving and modifying soil moisture drought and hydrological drought development. We give recommendations for robust drought definitions to clarify timescales of drought and prevent confusion with related terms such as water scarcity and overexploitation. Additionally, our understanding and analysis of drought need to move from single driver to multiple drivers and from uni-directional to multi-directional. We identify research gaps and propose analysis approaches on 1) drivers, 2) modifiers, 3) impacts, 4) feedbacks, and 5) changing baseline of drought in the Anthropocene. The most pressing research questions are related to the attribution of drought to its causes, to linking drought impacts to drought characteristics, and to societal adaptation and responses to drought. Example questions include: i) what are the dominant drivers of drought in different parts of the world?, ii) how do human





modifications of drought enhance or alleviate drought severity?, iii) how do impacts of drought depend on the physical characteristics of drought versus the vulnerability of people or the environment?, iv) to what extent are physical and human drought processes coupled, and can feedback loops be identified and altered to lessen or mitigate drought?, v) how should we adapt our drought analysis to accommodate 'changes in the norm' (i.e. what are considered normal conditions) over time? Answering these questions requires exploration of qualitative and quantitative data as well as mixed modelling approaches. The challenges related to drought research and management in the Anthropocene are not unique to drought, but do require urgent attention. We give recommendations drawn from the fields of flood research, ecology, water management, and water resources studies. The framework presented here provides a holistic view on drought in the Anthropocene, which will help improve management strategies for mitigating the severity and reducing the impacts of droughts in future.

**Keywords: Drought, Anthropocene, Drought definitions, Research framework**

## 1 Introduction

The hydrological system is intrinsically intertwined with the climate system, the environmental/ecological system and the social system (Fig. 1). These links are dynamic and interdependent. Natural water inflows and outflows vary and change in time and space, as do human water exploitation and associated activities, leading to what some have called a mutually co-evolving "hydrosocial cycle" (Linton and Budds, 2014, p.170). All these complex interlinked processes define the state of the hydrological system and the amount of water stored in the soil, groundwater, lakes, rivers and reservoirs. When there is (much) less water in the hydrological system than normal, as manifested in below-normal soil moisture levels, river discharge, groundwater and/or lake/reservoir levels, the system is perceived to be in drought, whether by natural causes (meteorological anomalies) or anthropogenic causes such as groundwater abstraction (Van Loon et al., 2016). Droughts can have severe consequences for water use in various sectors, for instance agriculture, drinking water supply and hydropower production, as well as having adverse impacts on ecosystems (Ciais et al., 2005; Lake, 2011; Sheffield et al., 2012; Grayson, 2013; Mosely, 2015; Stahl et al., 2015; 2016).

In recent decades, droughts have received increasing attention from policy makers and society, while drought research has made significant progress. Examples of this progress are: the continuous development of drought indices (Shukla and Wood, 2008; Bloomfield and Marchant, 2013; Stagge et al., 2015b); the improved understanding of the link between drought and atmospheric and ocean drivers (Fleig et al., 2010; Kingston et al., 2015); the influence of evapotranspiration (Teuling et al., 2013), snow (Staudinger et al., 2014) and geology (Stoelzle et al., 2014) on drought severity; drought monitoring and forecasting (Sheffield et al., 2014, Trambauer et al., 2015); and the effects of climate change on drought (Prudhomme et al., 2014; Trenberth et al., 2014; Wanders et al., 2015).

Still, many challenges remain. For example, the attribution of a groundwater or surface water deficit to its natural and human causes and the prediction of such a drought remain very difficult (Van Dijk et al., 2013; Diffenbaugh et al., 2015). For the



recent multi-year drought in California this has led to discussion about the role of groundwater abstraction (AghaKouchak et al., 2015a). Additionally, observed trends in measured low flows and drought are influenced by human activities (Sadri et al., 2016), probably even when only 'unregulated' catchments are selected (as noted by Hisdal et al., 2001; Stahl et al., 2010). This undermines our understanding of the effects of climate change on low flows and droughts and increases the uncertainty in projections for the future (Forzieri et al., 2014). Similar difficulties arise when attempting to link physical (i.e. climate or hydrological) indicators with societal or environmental impacts (Stanke et al., 2013; Bachmair et al., 2015a; Gudmundsson et al., 2014; Blauhut et al., 2015, Stagge et al., 2015a); this link being a crucial step in enabling societies to prepare for drought risks. In many big cities, for example, coping with drought is very complex, because vulnerability is high and factors such as the urban heat island effect, poor water supply, and water quality issues play an additional role (Güneralp et al., 2015). In drought management, the connections within the hydrological cycle are often overlooked, for example when unsustainable groundwater abstraction is used as adaptation to drought (e.g. Castle at al., 2014; Foster et al., 2015), or when restrictions are imposed for using surface water, but not for groundwater, leading to enhancement of the drought (as during the recent California drought and previous droughts in the Netherlands).

These examples of open questions point to gaps in our understanding of the complex interdisciplinary issue that is drought, and also highlight the unsuitability of current methods and data to answer these questions (see Box 1). For successful drought risk management our understanding must include the processes leading to drought (causes), and the impacts of drought (consequences). In this way drought predictions can be made and effective measure taken to mitigate drought severity and to reduce drought impacts.

The growing human impact on the earth system has led to numerous calls to recognise a new, distinct geological epoch, the 'Anthropocene'. While debate continues about the definition of the Anthropocene (Crutzen, 2002; Lewis and Maslin, 2015), it provides a useful framework for considering the present era, when human activity plays a fundamental role in water, energy and biogeochemical cycles. In the Anthropocene, society actively shapes water availability, and the feedbacks between physical and social aspects are particularly important during periods of water deficit. This means we cannot see drought as an external natural hazard and treat the consequences separately from the causes. Van Loon et al. (2016) argued that, for successful drought management in the Anthropocene, natural and human processes need to be fully integrated into drought definitions, process understanding, and analysis approaches. This paper builds on that argument and elaborates on research questions, data and methodology that are needed to reframe research in the Anthropocene.

## 2 Defining drought in the Anthropocene

It is known that human activities can create a drought situation or make an existing one worse (e.g. Wilhite and Glanz, 1985; Tallaksen and Van Lanen, 2004), but these processes are rarely ever explicitly included in drought definitions. Much has been said about the need for objective drought definitions and the difficulties related to that aim (e.g. Yevjevich, 1967; Wilhite and Glantz, 1985; Lloyd-Hughes, 2014), which we will not repeat here. We do, however, need to have a closer look





at identifying the role of human processes in the definition of drought. In this section, we therefore revisit drought definitions and make suggestions for robust use in the Anthropocene.

## 2.1 Drought as a lack of water

Drought is defined as a lack of water compared to normal conditions which can occur in different components of the hydrological cycle (Palmer, 1965; Tallaksen and Van Lanen, 2004; Sheffield and Wood, 2011). It is commonly subdivided into meteorological drought (rainfall deficit), soil moisture drought (below-normal soil moisture levels) and hydrological drought (below-normal (sub)surface water availability). The normal is often taken as a (percentile of the) climatology of the variable of interest, and severity (e.g. deficit volume) and duration of drought events can be calculated (Van Loon, 2015).

In the natural sciences, there is a fair understanding of the propagation of drought from meteorological drought to soil moisture drought and hydrological drought (Fig. 2 – left side), influenced by catchment properties such as geology and vegetation cover. For example, many hydrological drought types have been recognised, e.g. the classical rainfall-deficit drought, but also hydrological droughts caused by temperature anomalies in snow-dominated areas (Van Loon and Van Lanen, 2012; Van Loon et al. 2015). This is typically regarded as a uni-directional propagation with human receptors at the

downstream end. However, in reality, human processes are interlinked with natural processes in various ways (Fig. 2 – right side). Soil moisture and hydrological drought (hereafter called drought) are the result of low inputs to the hydrological system (e.g. lack of rain, snow/glacier melt, irrigation, sewage return flows), high outputs (e.g. evapotranspiration, human water use) and limited storage (in soil, groundwater, lakes and reservoirs). Human activities influence water input, output and storage and, therefore, modify the propagation of drought and can even be the cause of drought in the absence of natural

drivers of drought. The drought typology based on natural processes should therefore be complemented with drought types based on human processes.

The natural drought types can be grouped together as "climate-induced" droughts and drought types based on human processes can be termed "human-induced" or "man-made" drought (Fig. 3; Van Loon et al., 2016). This parallels an existing widely-referenced typology of floods, which includes "man-made flood" alongside natural floods such as flash flood,

snowmelt flood, and ice jam flood (e.g. Yevjevich, 1994; De Kraker, 2015). The distinction between climate-induced and human-induced drought is useful in studies of the attribution of drought to its causes. To further acknowledge the possibly large influence of human activities modifying drought (Fig. 2), we additionally propose the term "human-modified drought" for a drought that is enhanced or alleviated as the result of anthropogenic processes (Fig. 3). For this terminology, we focus on direct human influences on the hydrological cycle such as water abstraction and land use change, although we recognise

that anthropogenic climate change indirectly affects the meteorological drivers of drought (e.g. Williams et al., 2015).

With these terms, we actively include humans as drivers and modifiers of drought in the definition. There is no need for rephrasing the general drought definition, in which human processes are implicitly included. Furthermore, the terms we





propose are not new (climate-induced drought: Sheffield and Wood, 2011, p. 30; human-induced drought: Wilhite and Buchanan-Smith 2005, p. 10 and Falkenmark and Rockström, 2008, p. 93) and they match well with the flood terminology (Yevjevich, 1994).

## 2.2 "Drier than normal": timescales of drought in the Anthropocene

Drought is a lack of water compared to a certain 'normal situation', but what constitutes this normal situation in the Anthropocene? A drought occurs when actual water levels are below normal (Fig. 4). In a natural catchment, undisturbed by human activity, both actual and normal water levels are governed by natural processes in response to climate. Normal water levels are determined by the climate (long timescales), for example a (semi-)arid climate results in low average water levels. Actual levels are determined by climate variability (here used as term for a combination of weather events; short timescales),

for example a rainfall deficit leading to a climate-induced drought.

In a human-influenced catchment, actual and normal water levels are, besides by climate, also influenced by human activities. The actual situation is influenced by water use (short timescales), whereas the normal situation is influenced by long-term actions such as groundwater depletion and anthropogenic land use change (long timescales). For example, in the Jucar basin in Spain drought measures are based on thresholds in measured reservoir levels, groundwater levels, and river

flow, which are all heavily influenced by abstraction for irrigation (Andreu et al., 2009).

Because drought is an extreme event, the normal situation is not characterised by average water levels. Instead, a drought threshold (Fig. 3) is used that is calculated as a percentile(s) of a long time series (commonly, the value that is exceeded 80-95% of the time) or return periods representing rare occurrence (for example, a 50 year drought). Some studies use a variable threshold to represent seasonality and identify differences between droughts in different seasons (Van Loon, 2015). This is

very relevant in the Anthropocene, because humans interact differently with droughts in different seasons. Water abstraction for irrigation, for example, also follows a seasonal pattern and has different effects on summer drought vs. winter drought. On the other hand, in monsoon climates, drought characterised by a prolonged dry season causes different socio-economic impacts than a below-normal wet season.

## 2.3 Confusion between terms in the Anthropocene

Drought is often confused with water scarcity and water shortage, which are defined as 'less water than needed', i.e. where demand is greater than supply. The demand, or desired level, is included in Fig. 4 to illustrate the difference. In an unpopulated natural region, the desired situation is related to ecosystem requirements. Often these are not different from the normal situation because of the co-evolution of ecosystem and landscape. However, in a human-dominated region, the desired situation is related to water demand, which is dependent on population, standard of living, water efficiency, but also

on climate. In many areas the desired situation is out of balance with the normal situation because of rapid population growth, changes in diet, etc. This long-term imbalance leads to water scarcity and when it is complemented with short-term drought it leads to acute water shortage (Fig. 4; Table 1). If society satisfies its demand by abstracting more water, human-





induced drought can occur in the short term (changing the actual situation) and overexploitation in the long term (changing the normal situation; Table 1).

Human-induced drought should also not be confused with the term "socio-economic drought" (Wilhite and Glantz, 1985, p. 115), which is used to denote socio-economic impacts of drought. Although socio-economic drought is often mentioned as a type of drought in scientific papers and on websites explaining drought to the general public, a clear distinction should be made between the physical lack of water (drought) and its socio-economic consequences (impacts of drought). These impacts are sometimes used to define the drought threshold (Fig. 3), which then reflects the water level at which ecological or socio-economic impacts are expected to occur, such as ecological minimum flow or minimum reservoir levels.

We have to point out that the definitions of drought and its impacts used here deviate from the definitions used in other scientific disciplines, in particular in the climate community. In the IPCC SREX report the term "extreme (weather or climate) event" is used, having 'impacts' defined as the "spectrum of outcomes for humans, society, and physical systems, including ecosystems" (IPCC, 2012, p. 40). Drought, as we define it here, is then considered an "impact on the natural physical environment" (IPCC, 2012, p. 167). Similar confusion can arise for the terms 'attribution', 'mitigation' and 'adaptation', which are often assumed to be synonymous with attribution, mitigation and adaptation of (anthropogenic) climate change, but can also be used for the attribution, mitigation and adaptation of drought.

## 3 A framework for understanding and analysing drought in the Anthropocene

The traditional view of drought propagation is uni-directional: climate variability causes drought, which propagates through the hydrological system and subsequently leads to impacts (Fig. 2 – left side). Because of the complex relationships in the water cycle (Fig. 1) there are other drivers and modifications of drought and influences working in the opposite direction (Fig. 2). Therefore, the understanding of drought propagation needs to move from single driver to multiple drivers, and from uni-directional to bi-directional or even multi-directional.

For characterisation of this complete multi-directional system, unfortunately, our understanding and observation of drought processes have important gaps and the modelling and prediction tools at our disposal are therefore inadequate. The gaps are in the areas of 1) drivers of drought, 2) modifications of drought, 3) impacts of drought, 4) feedbacks of drought, and 5) changing norms. The framework presented in this section allows us to acknowledge what has been done in these areas, highlight where our understanding of drought processes in the Anthropocene is lacking and discuss the data, approaches and tools that are needed to address these gaps.

### 3.1 Drivers of drought in the Anthropocene

Drought is often seen from a meteorological perspective (Van Lanen et al., 2016), driven only by meteorological anomalies that disturb the normal water balance in a catchment (Fig. 2 – left side). Given the significant human modifications of the terrestrial hydrological cycle, this is too simplistic a perspective (Box 1). If we take a hydrological perspective on drought, a



lack of water compared to normal conditions can have a range of drivers (Fig. 2). There are many reasons for adopting a hydrological rather than meteorological perspective on drought. Firstly, people mainly use (sub)surface water, not rainfall directly (except for rainwater harvesting), so socio-economic impacts of drought are more related to a lack of (sub)surface water. Secondly, water on and beneath the land surface can be managed and manipulated, in contrast to rainfall, so that
hydrological drought can be mitigated. And finally, the direct anthropogenic influences on hydrological drought are probably much larger than climate change influences in many areas of the world. If we adopt a hydrological perspective on drought, it is important to distinguish between the different drivers of drought. This distinction leads to more accurate drought prediction and helps to direct attention and allocate investments between adaptation to climate-induced drought and reduction of human-induced drought. However, separating between climate-induced and human-induced drought is a major
scientific challenge.

Human-induced droughts are recognised (Wilhite and Buchanan-Smith, 2005), but there is a large gap in our understanding of the development of human-induced/-modified drought. We do know that human drivers principally influence soil moisture drought and hydrological drought and generally do not cause meteorological drought (Fig. 2; excluding relatively small scale land surface feedbacks, e.g. due to irrigation (Tuinenburg et al., 2014); or the global, indirect effects of
anthropogenic climate change). We can also hypothesise that the main process underlying human-induced and human-modified drought is abstraction from groundwater and surface water. There are many scientific studies on the long-term effects of abstraction (decades to centuries), but few on the temporal variability of abstraction on drought timescales (months to years). It is therefore still unclear how important human-induced and human-modified droughts are compared to climate-induced droughts for different areas around the world.

**Research questions** about drought drivers include: to what extent can observed historic drought events be attributed to different drivers? What are the dominant drivers of drought in different parts of the world? Do human-induced and human-modified droughts follow the same development as climate-induced drought and what are the implications for management? Answering these questions requires quantification of the direct human drivers of soil moisture drought and hydrological drought, in absence of meteorological anomalies, for historical drought events. The approach would be to identify droughts
in time series of observed hydrological variables and compare those to time series of climate-induced drought (represented by meteorological drought, observed droughts in an undisturbed nearby catchment, or simulated 'naturalised' droughts). This last approach was used successfully in Australia (Van Dijk et al., 2013) and Spain (Van Loon and Van Lanen, 2013) and could be applied in other areas around the world to understand the variability in how human drivers impact drought. Naturalisation of disturbed time series is challenging, being very much dependent on accurate modelling or regionalisation
approaches and data of human disturbances at a sufficiently high spatial and temporal resolution. Many international hydrological databases and data-sharing initiatives, however, have deliberately focused on near-natural systems (e.g. Hannah et al., 2011; Whitfield et al. 2012) in order to discern climate-driven processes from the noise of various human disturbances. We argue for more analysis of the disturbed catchments already included in hydrological databases and promote the extension of these databases with more human-influenced catchments, as suggested previously by Gustard et al. (2004).





Perhaps the greatest obstacle to achieving this is the lack of metadata indexing the type and degree of human impact in any one catchment, which is often not known or poorly quantified. There is a pressing need for a 'bottom-up' approach to transfer such knowledge, where it exists, from catchment, regional or national scale archives to the international research community. We also call for more experimental catchments in human-influenced areas in which particular human influences

on the hydrological cycle can be isolated and controlled, for example within the Euromediterranean Network of Experimental and Representative Basins (ERB), the network of Critical Zone Observatories in the USA, and the TERrestrial ENvironmental Observatories (TERENO) in Germany. Alternatively, we can make more use of satellite data of hydrological variables, which have become more widely available on global scale, although still with high uncertainties (AghaKouchak et al., 2015b). Useful satellite products are soil moisture missions (SMAP, SMOS, AMSR-E II, ASCAT) for soil moisture

information on high spatial and temporal resolution and NASA's Gravity Recovery and Climate Experiment (GRACE) for total water storage. If these are compared with global precipitation estimates (from satellites, TRMM and GPM, or from re-analysis), human-induced droughts might be identified in the absence of natural drought drivers.

### 3.2 Modifications of drought in the Anthropocene

The severity of droughts is strongly modified by catchment storage and release processes. In the natural situation these

modifiers are determined by factors such as soil type, geology, land cover (Fig. 2 – left side). In the Anthropocene, human activities change storage and land properties influencing propagation processes, and modify drought severity directly through anthropogenic inflows or outflows of water (Fig. 2 – right side). Just like natural modifiers, human modifiers can have both positive (enhancing) and negative (attenuating) effects on drought severity. The processes underlying direct modification of drought severity by human influenced inflows or outflows of water are most recognised and understood,

whereas the effects of human modification of storage and land properties, although recognized as potentially important, are more elusive.

There are ample examples of how human changes in land properties influence the hydrological cycle. Urbanisation for example results in less infiltration and more runoff in some cases and in more recharge in others (due to leakage of water supply and sewage systems; Lerner, 1990). Deforestation, afforestation, agricultural practices and desertification influence

evapotranspiration and consequently soil moisture. Some studies focused on the effects of land use change on low flows (Tallaksen, 1993; Hurkmans et al., 2009), but there is very little quantitative research on how these processes influence drought severity.

**Research questions** about human modifications of drought include: how do human modifications of drought enhance or alleviate drought severity? How do we predict drought development, severity and recovery in human-influenced areas,

taking into account relevant human drought modifiers?

Direct inflows or outflows of water are relatively easy to quantify with a water balance approach that explicitly takes into account human water flows (Lloyd-Hughes, 2014). However, this approach requires data of human influences on the water system, such as surface water and groundwater abstraction, interbasin water transfers, and irrigation return flows. These data





are usually not measured or collected, and if they are, there are often privacy issues in sharing the data, even for research. Additionally, there are many illegal or undocumented human influences on the water system that remain unknown (e.g. Pérez Blanco and Gómez, 2012). National statistical databases can be a good source of information, but their spatial resolution is often coarse so downscaling might be needed. Examples of methods for downscaling information on water

demand and water use can be found in Wada et al. (2011) and Nazemi and Wheater (2015a,b). More qualitative and local scale information on the human influences in a catchment can be gathered by a range of methods including interviews with local water users, participant diaries, oral recollections, community histories, participant observation, photographs and other visual materials, satellite-derived land use maps, and novel methods such as unmanned aerial vehicles (also known as drones).

Besides new data, new methods are needed to disentangle human modifiers from natural modifiers of drought and quantify how large their effect on drought severity has been for historical drought events and might be for future events. When sufficient data are available, statistical methods, such as multiple regression analysis, can be useful in finding the statistical relationships between drought severity and multiple influencing factors. This approach was used by Van Loon and Laaha (2015) for natural drought modifiers, but can easily be extended to include human modifiers. Paired catchment statistical

approaches (as applied to urbanisation impacts on floods by Prosdocimi et al., 2015) or upstream ('natural') – downstream ('disturbed') comparisons (Fig. 5a; López-Moreno et al., 2009; Rangecroft et al., 2016) are other data-driven approaches, although these have yet to be applied extensively for drought and low flows. Another large-scale data analysis method that has great potential for use in drought research is comparative analysis (Wagener et al., 2007) that aims to find patterns by analysing a large set of catchments with a wide range of characteristics, both in terms of natural and human processes. This

method is especially valuable if it is combined with qualitative data to explain the patterns found.

For scenario testing, conceptual models of human-water systems (Di Baldassarre et al., 2013; 2015) are a useful tool. Natural flows are altered by the presence of reservoirs and the resulting outflows depend on (changing) operational rules, i.e. optimised for flood or drought (Fig. 5b). The conceptual model (Martinez et al., 2016) simulates how the occurrence of a flood event might lead to changes in operational rules (e.g. shifting from the "optimised for drought" to "optimised for

flood" scenario in Fig. 5b), which will eventually enhance the next drought event (Di Baldassarre et al., 2016).

Modelling tools are also indispensable for prediction of drought under human modification. There are many types of models and many options to use these models for drought in the Anthropocene. Classic large-scale hydrological models are being adapted to include more anthropogenic processes (e.g. Wada et al., 2011; Döll et al., 2012; Nazemi and Wheater, 2015a; Veldkamp et al., 2015). Analysing these models specifically during drought periods has given some encouraging results (Fig.

5c; e.g. Van Lanen et al., 2004; Verbeiren et al., 2013; Wada et al., 2013; Forzieri et al., 2014; Wanders and Wada, 2015), although model uncertainties during low flow and drought remain high. Since many human influences on the hydrological cycle are on local scale, hyper-resolution modelling might be needed to explicitly represent all relevant human activities (Wood, et al., 2011). For parameterisation of these models, however, a thorough understanding of the processes is essential (Beven and Cloke, 2012). Once again, the key limiting factor is availability of data and metadata on the human modifiers. If



information on human pressures is available, modelling can be a key tool in separating human and natural drivers (thus paving the way to attribution) through a 'multiple working hypothesis' approach (see for example the work of Harrigan et al., 2014).

## 3.3 Impacts of drought in the Anthropocene

On the other side of the propagation diagram are the environmental and socio-economic impacts of drought (Fig. 2). Drought impacts, compared to the impacts of other hazards, are mostly non-structural and difficult to quantify. Drought impacts also have a high diversity, ranging across agriculture, water supply, industry, energy production, human health, aquatic ecology, forestry and other sectors (Stahl et al., 2016). Impacts are sometimes characterised into direct and indirect or tangible and intangible impacts (Wilhite and Vanyarko, 2000). Thus, the quantification of drought impacts depends on the affected sector

and on the level of impact (direct or indirect, and perhaps cumulative). Direct impacts on the agricultural sector are often documented as losses or reductions in crop yields. However, associating indirect economic losses directly to drought is not always straightforward (Ding et al., 2011). Indirect negative consequences are often quantified by the number of people affected or by number of people who died as a result of related food security or health issues, but other factors than a direct association to drought may play an important role as well. Especially drought impacts on (mental) health are complex and

dependent on a multitude of factors (Stanke et al., 2013; Obrien et al., 2014).

Whether a drought event has negative consequences on one of these sectors also depends strongly on people's perception and thus on the vulnerability of affected sectors (Knutson et al., 1998; Iglesias et al., 2009). Understanding a particular sector's vulnerability can benefit from specific information and quantification of drought impacts in addition to knowledge on the general vulnerability factors that describe the sensitivity and adaptive capacity of the considered community or region.

For drought characteristics, ample data sources exist. However, as noted before, they rarely specify the level of human modification to the drought signal. For vulnerability analysis, many useful data on the sensitivity or adaptive capacity from community to country to international levels are lacking (De Stefano et al., 2015). For drought impacts, the US Drought Impact Reporter (DIR) (http://droughtreporter.unl.edu/) and the European Drought Impact report Inventory (EDII) in Europe (http://www.geo.uio.no/edc/droughtdb/) collect and categorise textual drought impact reports, whereas Lackstrom et al.

(2013) and others suggest the development of a more targeted impact monitoring.

**Research questions** that need to be addressed thus include: how should drought impacts be monitored and quantified? How do they depend on the physical characteristics of drought versus the vulnerability of people or the environment?

Retrospective analysis of the physical characteristics of past droughts (through some drought indicator) and the impacts that they have triggered is one way forward, if compared across different societal contexts, in particular different degrees of

vulnerability. However, methods to link physical indicators and societal impacts have only recently been explored more in-depth. Figure 6 gives an overview of the different methods. The most widely adopted approach to relate drought indicators-to-impacts is to link commonly used hydrometeorological drought indicators to agricultural yield (Lobell et al., 2008; Vicente-Serrano et al., 2012; 2013; Bachmair et al., 2016). Most of these studies are based on correlation and as summarized





by Stagge et al. (2015a), thus are useful for screening relationships, but they measure the response of a variable, such as crop yield, across its entire range of values including typical or even productive years. A further complicating factor is the non-linearity of the climate-yield relation, which can show ambiguous relations with positive effects during drought or threshold behaviour for reductions in yield (Fig. 6a). Report-based impact data cover a wider range of impact types, but are tedious to

gather and have many biases. So far they have mostly been converted to binary or counts of "impact occurrences" for indicator-to-impact studies (Fig. 6b). Data-driven statistical models have used time series or spatial variability of these "impact occurrences" as a response variable in regression and classification tree models (Fig. 6c; Stagge et al., 2015a; Bachmair et al., 2015; Blauhut et al., 2015). These studies have also shown that impact generation is more complex than previously assumed and may be caused by the co-occurrence of several extremes, lagged effects, and seasonality (Stagge et

al. 2015a). A useful outcome of these modelling exercises was the objective determination of 'best-indicators' for impacts in particular sectors that are strongly influenced by human factors. For example, when using the Standardized Precipitation Index (SPI) or Standardized Precipitation Evapotranspiration Index (SPEI) the best accumulation period suited to predict agricultural impacts clearly differs for irrigated and rain-fed agriculture (Stagge et al., 2015a); similarly, the best accumulation period to predict drought impact on public water supply differs depending on the relative contributions of

groundwater versus surface water resources and the type of reservoirs available (Bachmair et al., 2015b). These examples show that human perception of drought impacts can differ from the occurrence of drought in the natural hydrological system, depending on the prevailing water management framework. Future analysis could use impact information to better characterise impacts of human-modified or human-induced drought.

### 3.4 Human feedbacks of drought in the Anthropocene

The interaction between natural hydroclimatological processes and human influences is not a simple addition of both effects, but instead comprises complex and dynamic feedbacks resulting in a strongly non-linear response of the hydrological system (Fig. 2). There are negative feedbacks, when human management responses to drought (impacts) lessen drought; and positive feedbacks, where management responses exacerbate drought. There is growing knowledge of climate feedbacks (also called land-atmosphere feedbacks), in which drought influences evapotranspiration rates positively or negatively

(Teuling et al., 2013), dependent on geographic situation and time frame. There is, however, only very limited understanding of human feedbacks during drought.

Short-term human feedbacks are responses to drought situations (whether observed, or at least perceived, or predicted) that influence water storages and fluxes within a particular water system in a catchment over timescales of days to years. These influences can include reductions in water use, implementation of water saving technologies, planting of less water-

demanding crops, using other water sources (e.g. from surface water to groundwater; from clean to grey water), short-term increases in groundwater abstraction because of surface water shortage, and water transfer from wetter areas, or areas where water demand is lower (e.g. Andreu et al., 2005).





There is strong non-linearity in the reaction of the water system to these short-term influences (Sivapalan et al., 2012). Timescales often do not match; for example, the societal response might be in the order of weeks, but the reaction of groundwater can be in the order of years (Gleeson et al., 2010; Castle et al., 2014). Consequently, there is a difference between short- and long-term droughts, where longer droughts show a more complex interaction of natural and human processes (Van Dijk et al., 2013). Societies can also learn from historic droughts and adapt drought policy in the long-term to be more pro-active, rather than reactive, when the next drought comes (McLeman et al., 2014). Crucially however, human activities are not only influenced by climate and the drought state of the system, but are also strongly dependent on domestic water behaviours (Pullinger et al., 2013), national policy styles (Gober, 2013), existing public policies (particularly for agriculture; Campos, 2015), water law and governance (Maggioni, 2015), and even indirectly by international food markets and geopolitics.

**Research questions** about human feedbacks include: are there commonalities in the response of different societies to different drought events? To what extent are physical and human drought processes coupled, and can feedback loops be identified and altered to lessen or mitigate drought? What are the links between discourses and practices of drought mitigation and alleviation? Additionally, more information is needed on past histories of water use and the role of technology in current routines of water practice (Pullinger et al., 2013), tipping points in human water use (Mera et al., 2014), and the reasons for a lack of public awareness of environmental water demands (Dessai and Sims, 2010). Understanding the relationship between these factors is crucial to enhancing our understanding of drought.

Qualitative data are essential in our quest for increased understanding of this topic. One novel type of qualitative data is the use of drought narratives, which can give new insights into societal responses and feedbacks. This is an example of how citizen science can help harvest data. It is especially interesting to study 'paired drought events', i.e. drought events of similar magnitude that occurred in the same region, to investigate whether societies learn from drought events and what the effect of this learning is on the next drought. Despite the obvious uncertainties of such an approach, it can provide information on drought responses and feedbacks from one drought event to the next, as was shown for 'paired flood events' by Kreibich et al. (in prep).

For quantitative prediction of the effect of feedbacks on drought, classic water management models could be adapted to include more hydrology and feedbacks. The modelling tools that are used in water management generally take water availability as external forcing and do not include the feedbacks of the water use on the hydrological system (e.g. Higgins et al., 2008; Borgomeo et al., 2014). Many water management models, however, are capable of simulating the effect of the allocation of water on hydrological processes also during drought, as was shown by Querner et al. (2008) and Van Oel et al. (2012), or simulating the influence of water management decisions on the evolution of a given drought scenario (e.g. Watts et al. 2012).

Socio-hydrology models aim to account explicitly for the two-way feedbacks between social and hydrological processes (e.g. Sivapalan et al., 2012). Di Baldassarre et al. (2013; 2015) have applied this approach to flooding, and the development of a similar modelling framework for drought is underway (Kuil et al., 2015). As the interplay between water and people is





still poorly understood, socio-hydrological theory is still to be developed via an iterative process of empirical study, comparative analysis and process-based modelling. Thus, while the current studies do contribute to improve the current understudying of water-society interactions, their predictive power is still very limited (Viglione et al., 2014). Modelling approaches are most successful when people themselves are actively involved in the modelling process; stakeholders can, for

example, guide scenario-analysis (Loucks, 2015). In contrast to modelling studies, environmental social science epistemologies, such as grounded theory building, offer alternative means of understanding water resource use and human behaviour (Pearce et al., 2013), potentially enabling more holistic insights into the role of drought feedbacks in the "hydrosocial cycle" (Linton and Budds, 2014, p.170).

## 3.3 Changing norm in the Anthropocene

We now live in a fast-changing environment; both climate change and long-term human influences on the water cycle are changing the norm, even within 30 year time blocks that are traditionally being used to determine a climatology or a drought threshold. This is important from a drought perspective because the normal situation is our reference to determine the occurrence and severity of drought events (Fig. 4). There are many uncertainties in dealing with extreme events like drought under conditions of change. Some model studies of future hydrological drought commented on the assumption of using the

same threshold for the historic and the future period (e.g. Giuntoli et al., 2015; Wanders et al., 2015). Two aspects should be mentioned. Firstly, regime changes trigger methodological considerations, because they can result in detection of drought events that should otherwise not be classified as drought, such as earlier snowmelt resulting in a 'drought' in the normal snowmelt period (Lehner et al., 2006; Van Huijgevoort et al., 2014). Secondly, ecological and societal systems might adapt to a changing norm, but it is unclear how fast these adaptations will take place and whether tipping points will be passed

(Mera et al., 2014).

**Research questions** related to changing norms include: is the norm actually changing or do we not have the data or understanding of natural variability to say anything about what is normal? How do long-term human influences on the water cycle change the norm? Do societies adapt to changes in the norm so that more severe droughts might lead to less impact in the future? How should we adapt our drought analysis to accommodate changes in the norm?

The most straightforward solution to regime shifts is analysing different seasons separately, as was done by Hisdal et al. (2001) and Feyen and Dankers (2009) with respect to a snow season and non-snow season. In historical drought analyses, long-term climate change effects are often excluded by taking a short enough period to neglect climate change or by detrending the time series. For a changing norm due to future climate change, Vidal et al. (2012) and Wanders et al. (2015) have suggested to include adaptation by changing the drought threshold for the future. Mondal and Mujumdar (2015)

followed a similar approach by estimating changes in return levels of drought under similar probability of occurrence in observed and projected streamflow. These methodologies should be evaluated more thoroughly and should also be applied to account for long-term human influences, alongside climate change effects. Important long-term human influences to





consider are anthropogenic land use change (urbanisation and deforestation; Verbeiren et al., 2016), continuous increases in abstraction, and step-changes in storage by dam building.

These methodological explorations on how to deal with changes of the norm in drought analysis are urgently needed, but we should also get a better understanding of long-term changes in the perception of drought impacts and vulnerability. This

perception drives adaptation to extreme events like drought and influences feedbacks between the physical and social system. Societies might be able to adapt to a changing mean, but they are more likely to be triggered by extreme impacts of a severe drought, resulting in long-term adaptations aiming to reduce impacts of drought in the future (Fig. 7; Smit et al., 2000; Dillehay and Kolata, 2004). More research is needed to understand trajectories of social development that lead to adaptation to drought.

We can benefit from the work done on long timescales, both regarding long-term climate change, long-term human influence on the water cycle (overexploitation) and long-term water demand and scarcity (Table 1).  Research on groundwater depletion (Aeschbach-Hertig and Gleeson, 2012) and water scarcity (Rijsberman, 2006) has been carried out on large temporal and spatial scales (annual and country level), because that is the level of relevance and the level of available data. Accounting for temporal variability and increasing spatial resolution can close the gap with drought research (Savenije,

2000; Hoekstra et al., 2012; Hering et al., 2015; Vörösmarty et al., 2015). Veldkamp et al. (2015) and Mekonnen and Hoekstra (2016) were the first to explore sub-annual time scales of water scarcity.

## 4 A broader scope on drought in the Anthropocene

The framework proposed here is in line with suggestions for hydrological research in general, for example with the call by Wagener et al. (2010) for a paradigm shift to study hydrology under change, with the research agenda set by Thompson et al.

(2013) for hydrological prediction in the Anthropocene, with the new decade of the International Association of Hydrological Sciences (IAHS) 'Panta Rhei' (Montanari et al., 2013; McMillan et al., 2016), and with the propositions for hydrological research and water management by Vogel et al. (2015). Complementary to these visions on the future of hydrology in general, we think that a focus on drought is needed to cope with complex future water challenges.

The challenges mentioned here are, however, not unique to drought. We can learn from other fields that have struggled or

are still struggling with similar issues. The parallels with flood research have already been mentioned above in relation to definitions and socio-hydrology. Flood research is further advanced than drought research in including human influences on catchments and rivers in flood analysis (e.g. Vorogushyn and Merz, 2013) and many studies exist that focus on attribution of flood to different drivers and modifications, the complex interaction between natural and human processes, and flood response and adaptation.

There is also an interesting parallel between society and ecology, because, just like people, plants are simultaneously dependent on and shape water availability (e.g. Rodriguez-Iturbe, 2001).  The field of ecohydrology has evolved in the last 15 years to a quantitative understanding of the interrelated dynamics of plants and water (e.g. Hannah et al., 2007;





Asbjornsen et al., 2011; Jenerette et al., 2012). The importance of including vegetation feedbacks in future drought modelling was, for example, highlighted by Prudhomme et al. (2014). Similar approaches can be applied to the interrelated dynamics of people and water, especially during drought. In addition, the field of hydroecology has been grappling for several decades with the same issue of how to capture 'reference' natural conditions in order to compare impacted conditions

against. Again, this is hampered because there are so few extant examples of natural conditions in observed hydrological datasets; the same challenges of how to naturalise flows have been at the core of the environmental flow paradigm (e.g. Acreman and Dunbar, 2004).

Societies have always had to cope with drought, so water management and governance have a long history. Especially interesting are the stories of civilisations that collapsed due to a combination of water overexploitation, drought and other

factors (e.g. Lucero, 2002). But there are many examples of successful water management in the past that have reduced drought severity or led to successful adaptation (e.g. Dillehay and Kolata, 2004; Garnier, 2015), which can help to understand feedbacks between society and the water system. In this light it is also very informative to understand how people deal with uncertainties in drought prediction (Kasprzyk et al., 2009; Wagener et al., 2010), which are partly caused by the gaps in our understanding and unsuitability of data and tools to quantify the interaction between people and drought in the

Anthropocene (Vogel et al., 2015). The use of drought predictions by society plays an important role in the impacts and feedbacks of drought.

Although water scarcity is very different from drought, and water demand is not the focus of this article, regions with high water demand often influence the water cycle more drastically, possibly resulting in more human-induced drought and human-modified drought compared to regions with low water demand. Additionally, high-demand regions will be more

severely impacted by drought than low-demand regions. Since increases in global water demand are projected for the future, enhancing water scarcity, collaboration between drought research and water scarcity research is urgently needed.

In focussing on human aspects of drought we should not forget the other parts of the complex interlinked system (Fig. 1). Ecological and environmental requirements are recognised but are often neglected during drought (Vörösmarty et al., 2010). For example, in the Murray-Darling Basin (Australia) water management mitigated the water supply and economic impacts

of drought, but at the same time strongly amplified the negative environmental impacts of drought (Van Dijk et al., 2008). Deterioration of water quality during drought can mean that water is available but cannot be used, for example due to algal blooms or salt water intrusion in deltas (Van Vliet and Zwolsman, 2008). Although water quality was not discussed in this article, we stress that there are many challenges related to water quality and drought in the Anthropocene that require further research (e.g. Mosely, 2015).

In this opinion article we have argued that drought in the Anthropocene is not an external natural hazard. Instead, the natural hazard is intertwined with human influences on the water cycle and feedbacks of society on drought. We, therefore, explicitly include human processes in drought definitions and clarify previous confusion with related terms such as water scarcity. We present a multi-driver and multi-directional drought framework, in which human drivers, modifications, impacts, feedbacks and changing norms of drought are included in drought research. This framework highlights gaps in our





understanding and indicates the tools and data needed. The elements of the framework have increasing complexity, from relatively straightforward aspects like human drivers and modifications of drought, to the more complex impacts of drought, to compound feedbacks and changing norms that integrate across all other elements.

The framework can be used to focus on a specific point or research question with the aim to solve part of the puzzle, or to
study the entire interrelated system with the aim to put the pieces of the puzzle together. In the end both approaches will hopefully result in a more holistic view of drought in the Anthropocene and consequently better drought management, in which the appropriate understanding and data and tools are used to take effective measures to mitigate drought severity, and to reduce drought impacts in the Anthropocene (Van Loon et al., 2016). This is of crucial importance now that the world is facing increasing human influence on the hydrological system, increasing dependence of society on water availability,
combined with significant population growth, and climate change possibly leading to an increasing frequency of extreme hydroclimatological events (Vörösmarty et al., 2000; Oki and Kanae, 2006).

## Author contribution

A. Van Loon initialised the ideas presented in this paper with H. Van Lanen, T. Gleeson, R. Uijlenhoet and A. Teuling. All authors contributed to the discussions that shaped the paper. A. Van Loon prepared the manuscript with parts written by J.
Clark and K. Stahl, and contributions from all co-authors. Figures were prepared by A. Van Loon, S. Rangecroft, G. Di Baldassarre, N. Wanders, K. Stahl, B. Verbeiren, and T. Gleeson.

## Acknowledgements

The present work was (partially) developed within the framework of the Panta Rhei Research Initiative of the International Association of Hydrological Sciences (IAHS). It draws from discussion in (amongst others) the EU FP7 Project
DROUGHT-R&SPI (282769), supports the work of the UNESCO-IHP VIII FRIEND-Water programme and is partly funded by the Dutch NWO Rubicon project 'Adding the human dimension to drought' (reference number: 2004/08338/ALW).

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



**Box 1: Examples of why humans are integral to drought and we should not focus on natural drought causes only.**

*Drought management: Rhine (the Netherlands)*

In the Netherlands, drought management measures and drought committee meetings start when the discharge of the river Rhine falls below a pre-defined level, independent of possible causes of the low river flows (e.g. lack of rainfall, lack of snow melt, Germany abstracting more water, etc.; Rijkswaterstaat, 2015).

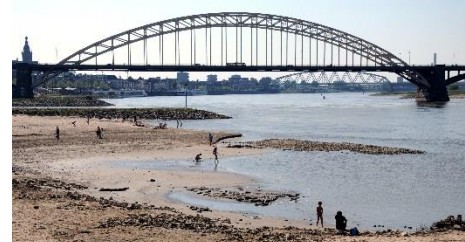

**Figure B 1:** 2011 drought on the River Rhine near Nijmegen (photo Ronald Puma; ronaldpuma.nl)

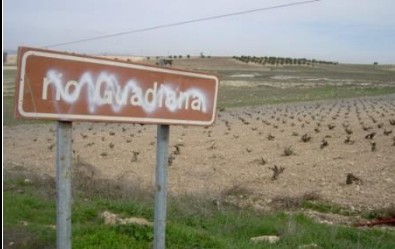

**Figure B 2:** Name of River Guadiana crossed out after being dry for 20 years.

*Drought attribution: Upper-Guadiana (Spain)*

An important wetland dried up in Spain in the 1990's. Nature organisations blamed the abstraction for irrigation by farmers, but farmers pointed to the severe multi-year lack of rainfall. There was a need to attribute the low water levels to their causes. Modelling showed that both parties were right, but that abstraction had four times as much influence than the lack of rainfall (Van Loon and Van Lanen, 2013).

*Drought termination: California (USA)*

"How much rainfall is needed to end the drought?" This question was and still is often mentioned in the media in California. We can calculate how much rain is needed to fill up the system, but at the same time we are constantly taking water out (for example by groundwater abstraction) and putting water in (for example by water transfers). Those human inputs and outputs cannot be disregarded in the calculation of how much rain is needed to end the drought.

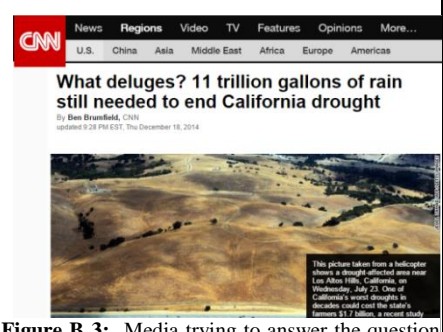

**Figure B 3:** Media trying to answer the question how much rain is needed to end California drought (from edition.cnn.com/2014/12/18/us/california-rains-and-drought, last access 13 May 2016)

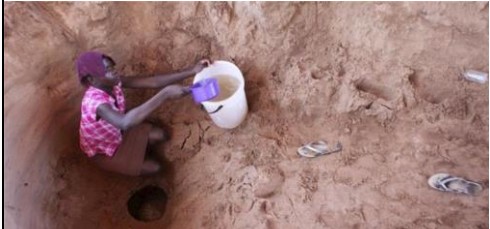

**Figure B 4:** Difficulty of water access in Africa (from www.ghanaweb.com/GhanaHomePage/world/South-Africa-won-t-declare-national-disaster-over-drought-413602, last access 13 May 2016)

*Drought impacts: Africa*

The impacts of drought are not only related to the severity of drought, but also to access to water sources, and possibility of using alternative sources. Most communities in Africa are very dependent on rain water and do not have access to alternative sources such as groundwater. A lack of rain then leads to severe impacts, even though groundwater reserves and nearby river basins might not suffer from drought (yet).



**Table 1: Drought terminology in relation to drivers and timescales**

| lack of water | | short term | long term |
|---|---|---|---|
| **compared to normal level** | **natural causes** | *climate-induced drought* | aridity |
| | **human causes** | *human-induced drought* | overexploitation |
| **compared to desired level** | | acute water shortage | water scarcity |




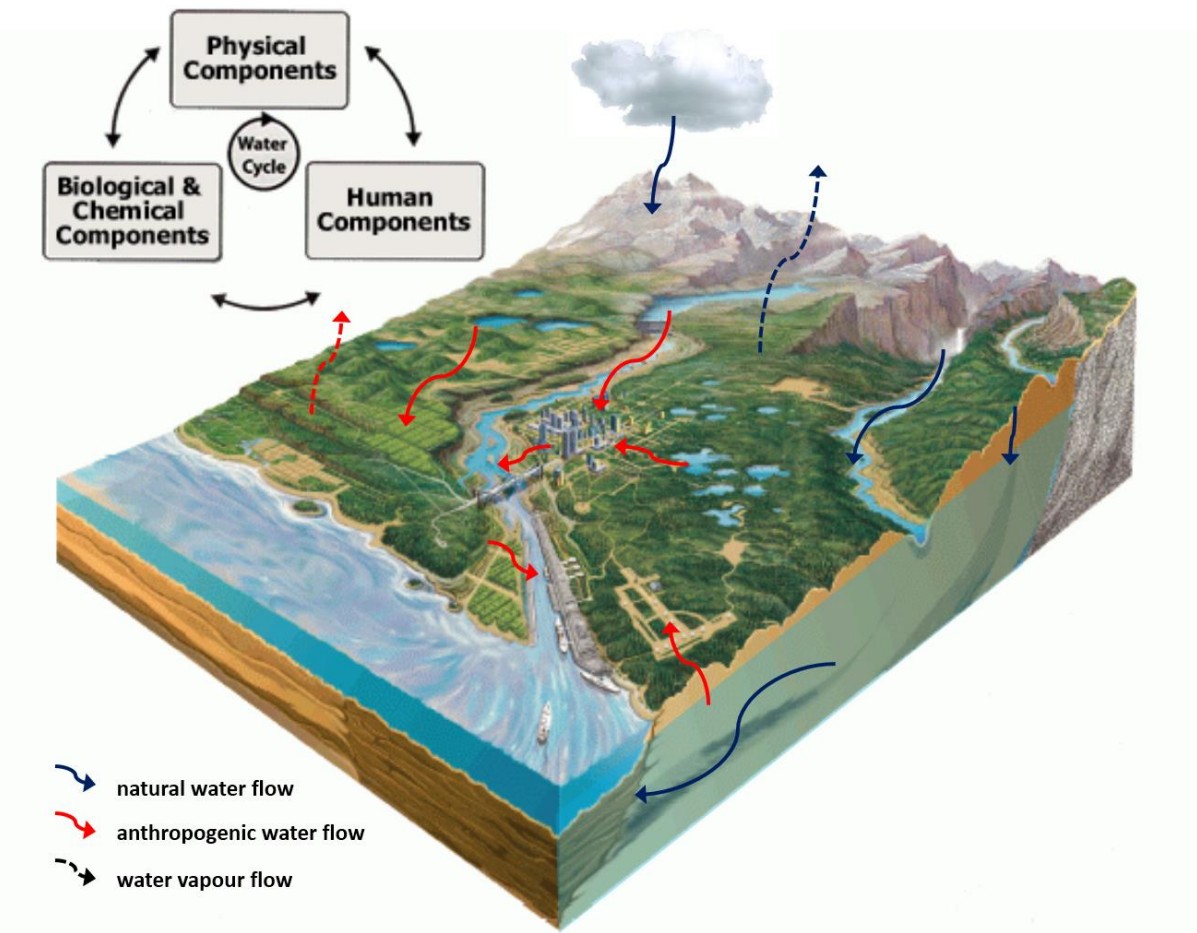

**Figure 1: The water system linking physical, biological and human components through natural and anthropogenic water flows (adapted from: Winter et al., 1998; Vörösmarty et al., 2004, copyright AGU).**




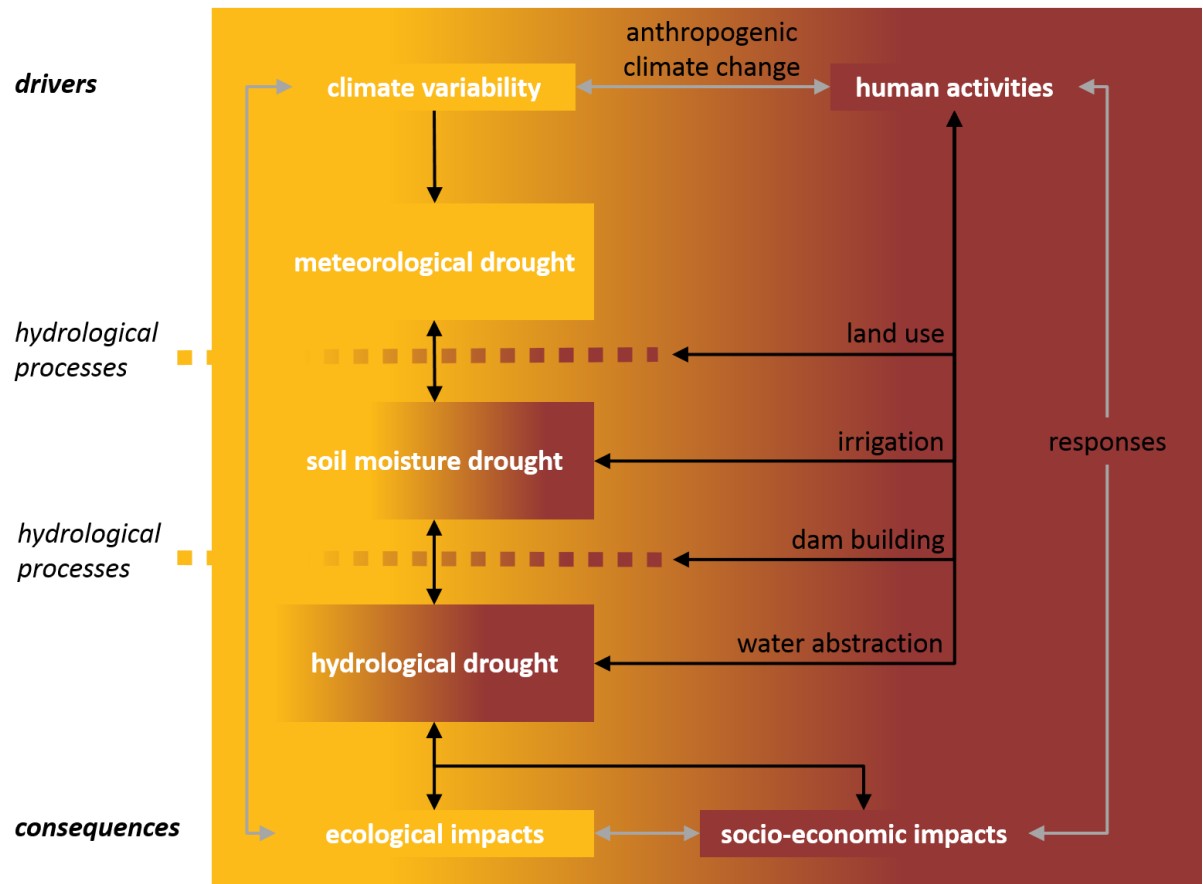

**Figure 2: Drought propagation including natural and human drivers and feedbacks, black arrows indicate direct influences, grey arrows feedbacks (modified from Van Loon et al. 2016).**

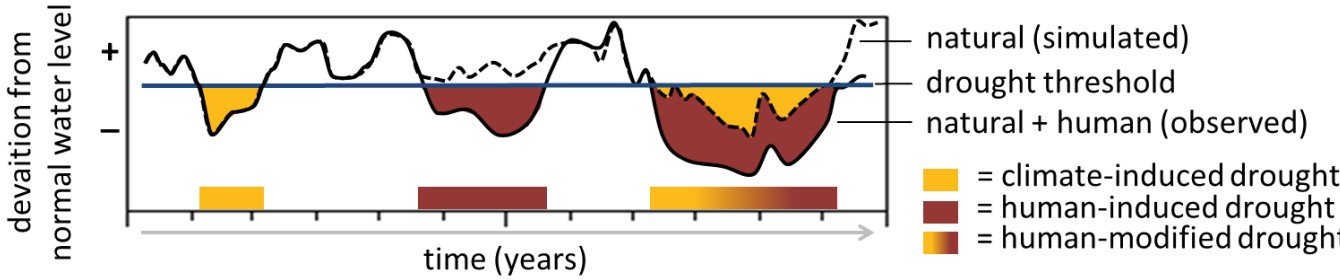

**Figure 3: Drought types: climate-induced drought, human-induced drought, human-modified drought (modified from Van Loon et al. 2016).**





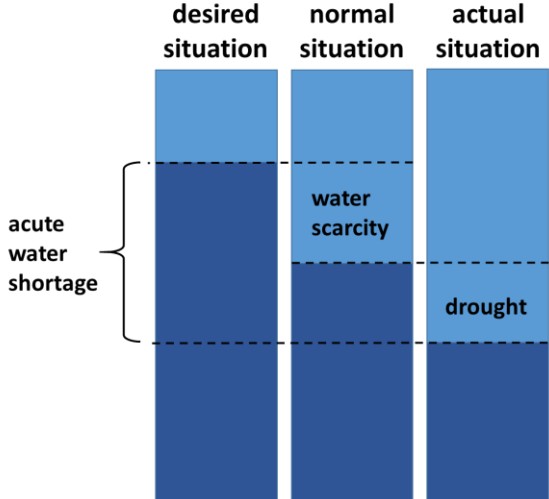

**Figure 4: Water storage (in river, lake, reservoir, or groundwater) on different timescales. If actual levels are below normal (low) levels, the hydrological system is in drought. If normal levels are below desired levels (determined by water demand), there is long-term water scarcity. If actual levels are below desired levels, there is acute water shortage. N.B.: water scarcity occurs on longer timescales than drought.**

(a)

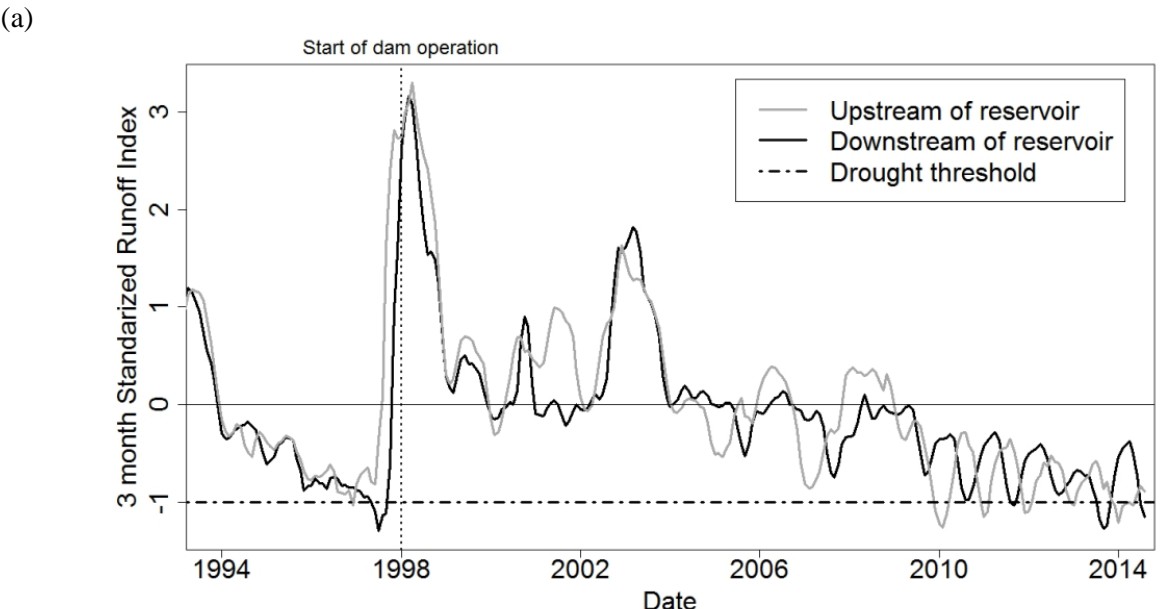





(b)

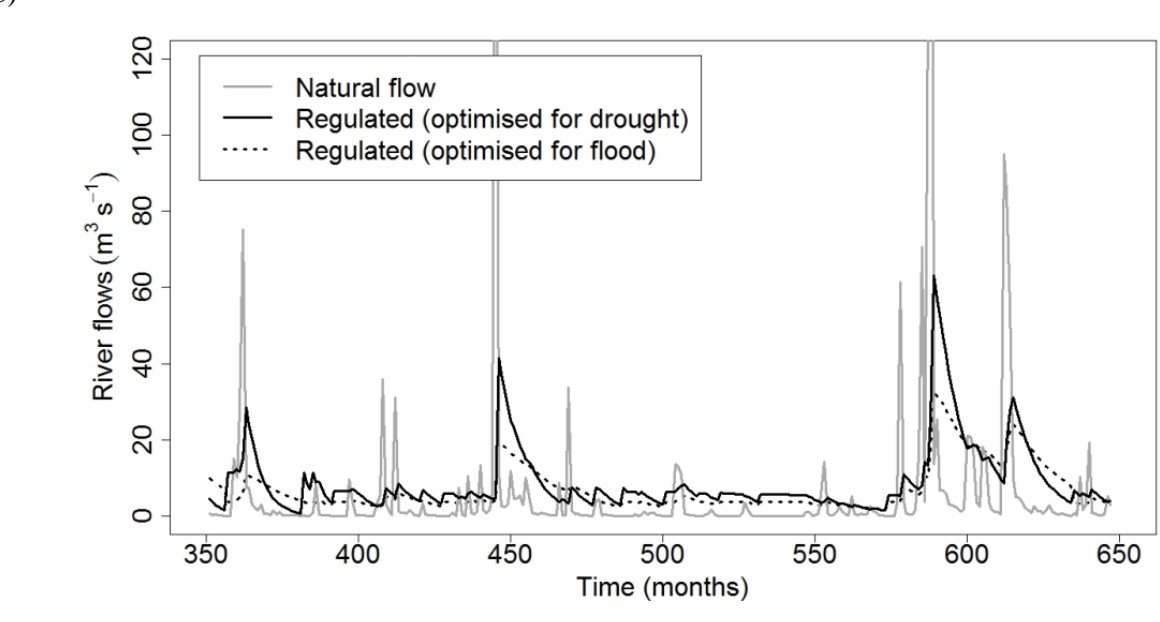

(c)

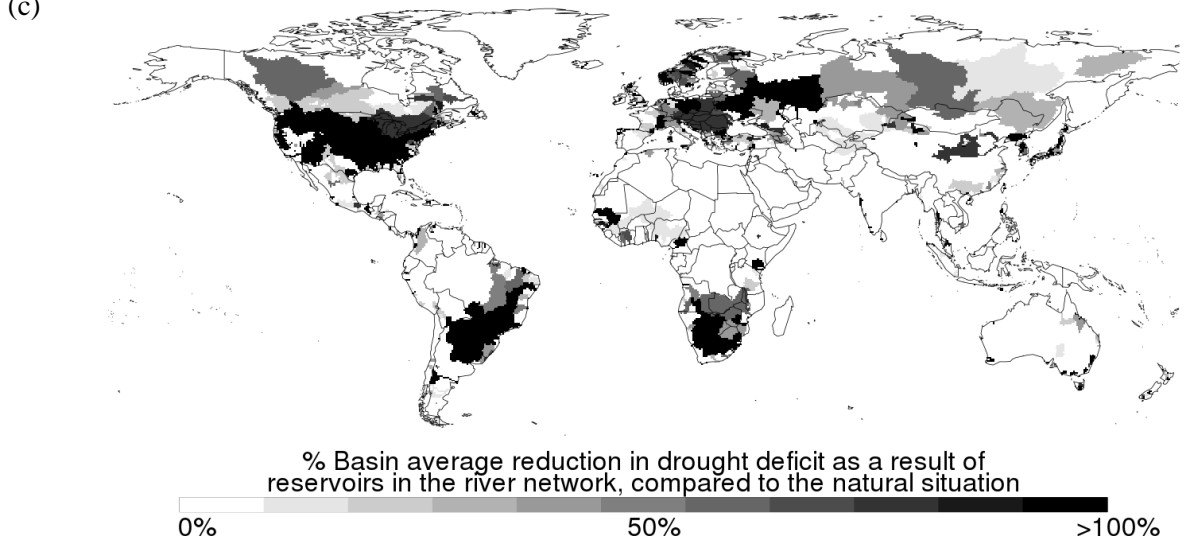

**Figure 5: Example of the approaches to investigate drought modification by reservoirs, a) based on observations of discharge upstream and downstream of a reservoir in Chile (Rangecroft et al., 2016), b) theoretical effect of reservoirs on drought (Martinez et al., 2016), c) simulated effect of reservoirs on drought deficit on global scale (adapted from Wanders and Wada, 2015).**



(a)                                    (b)                                    (c)

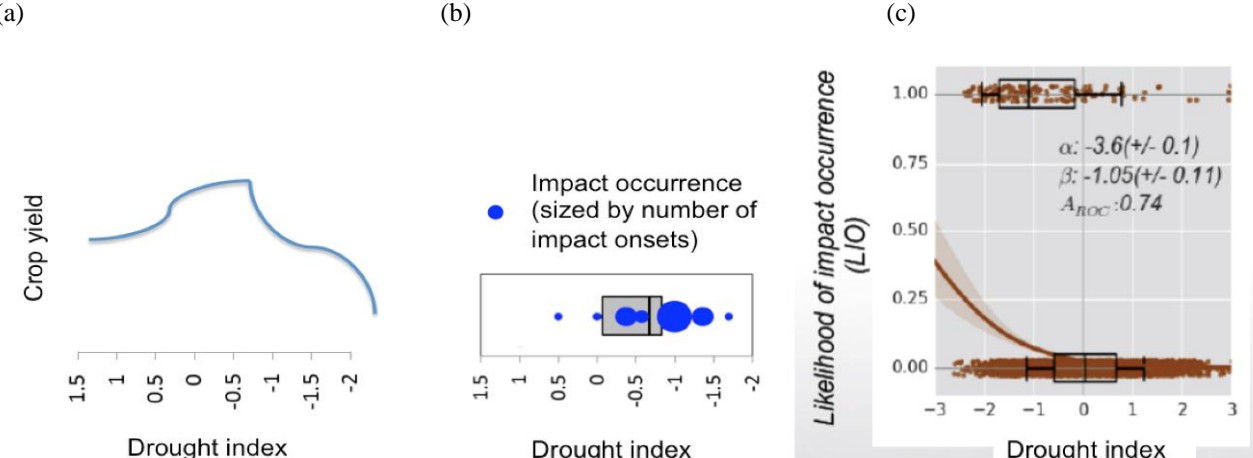

**Figure 6: Scheme of the three approaches to investigate impact-related drought index values: a) correlation of drought index to crop yield (from unpublished work), b) drought index values at the time of impact occurrence (based on Bachmair et al., 2015a) and c) logistic regression model predicting the likelihood of impact occurrence by the drought index (from Blauhut et al., 2015).**

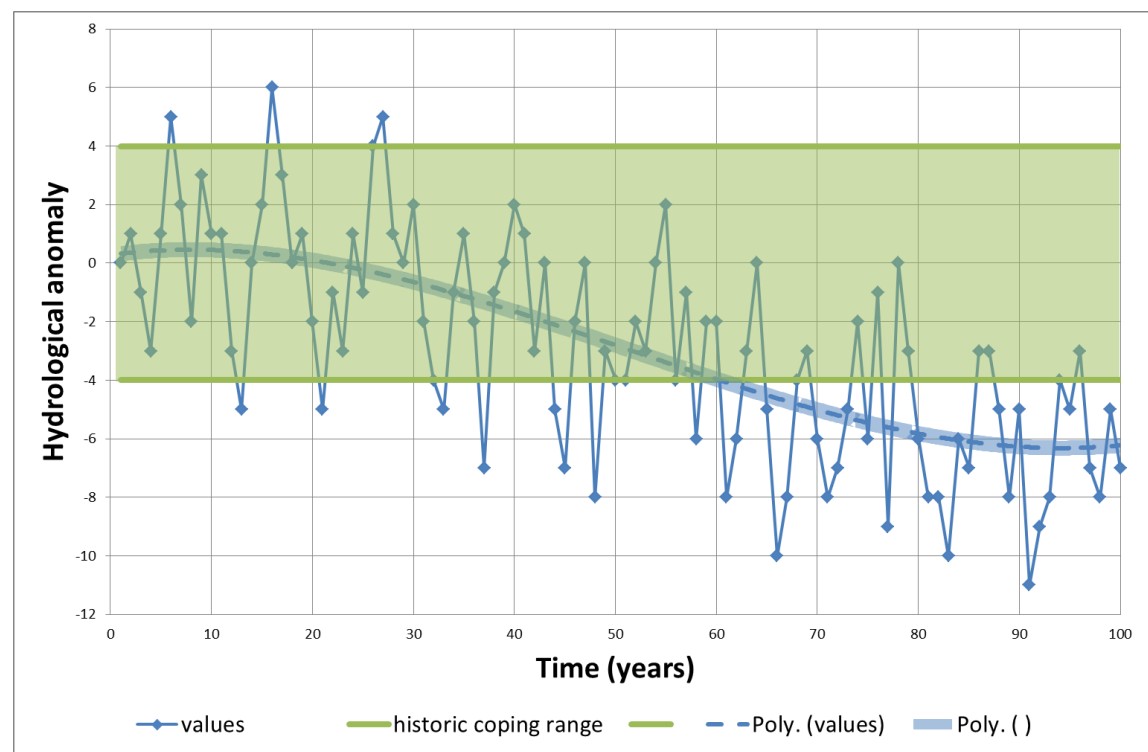

**Figure 7: Changing norm due to climate change changes drought occurrence and severity (after Smit et al., 2000). Will society adapt to changing norm or in response to one/two extreme events?**