# Peer review of "Drought in a human-modified world: reframing drought definitions, understanding and analysis approaches"

_Hydrology and Earth System Sciences, 2016_

## Short Comment (SC1) · 10 Jun 2016

I thoroughly enjoyed reading this paper and subscribe to most of its views. Indeed, it is time that we start including the human dimension into drought analyses and prediction. The paper sets up a comprehensive framework for defining the various types of drought in the Anthropocene and identifies a number of research gaps. There is one major point I would like to make about the framing of the paper. It is claimed on page 3, line 14 that there are "gaps in our understanding of the complex interdisciplinary issue that is drought". This seems to suggest that we as a community of hydrologist are unaware of the fact that drought is a many-headed dragon. I beg to differ here. I am not sure it is lack of understanding per se. It depends on who the "we" are in "our understanding".

[Figure]

Surely, hydrologists that are used to simulating or studying the hydrological system holistically, understand how the interactions between the hydrological states and fluxes and human intervention on these work conjunctively. It seems to me that it is more the general public and the policy makers that don't know or understand, partly because we as scientists lack a proper description of the interacting parts (ontology) or use the incorrect terms (semantics). So the question is not so much as "do we (as hydrologists) lack understanding?" but rather "do we as scientists lack the correct ontology and terminology for drought in the Anthropocene and therefore fail to convey understanding with the general public and the policy makers?".

This being said, this does not mean that all issues are resolved. Indeed, if we acknowledge that the study of drought also encompasses the human dimension and we also extend drought impacts from mere indices or water volumes to socio-economic and ecological impacts and if we want to understand how human water demand reacts to extreme events or a changing normal in the future, we need additional knowledge, i.e. understanding of additional system parts. In conclusion: I would set the stage a bit differently: there are two problems with current treatment of drought:

1) Lacking the correct ontology and associated semantics to describe what makes a drought, which results in lack of understanding of (the causes of) drought of public and policy makers.

2) Necessary extension of the drought research into including human impacts on drought, human response to droughts and the socio-economic en ecological effects on drought that leads to a number (you name 5) areas with lack of understanding of the (inner workings of) parts of the system.

Some minor points to make:

- Page 3, line 12: "Hydrological drought?

- Page 4, lines 26-28: Great definition. But what if human activities lead to reduced
drought? Think about the impacts on dams on low flows (your Figure 5) or return flows from deep (industrial) groundwater abstractions that also increased discharge in low flow periods (see de Graaf et al., Advances in Water Resources 2013)

-Page 5, line 30: Here there is a problem with your definitions. Before you defined "normal" as long term average under human and natural conditions. You should then add to the conditions that define a normal that there is long-term balance of fluxes.

- Page 12, line 24: remove the reference or find another. It is of no use to refer to an in prep paper.

- Page 12, lines 28-31 "Many water management. . ...scenario (Watts et al., 2012)". Actually that is also the case for many of the global hydrology and water resources models such as PCR-GLOBWB where water abstractions are a function of water availability".

- Page 13: de section title "Changing norm in the Anthropocene". Given the terminology you use, it is not better to use "changing normal" (see also your Figure 4 and the use of the term "normal" in meteorology)? A norm has also a normative connotation.

---

## Author Comment (AC1) · 29 Jun 2016

Thanks for your positive evaluation of our paper. In this reply we will shortly discuss your comments. A more detailed reply will follow with the revised version of our paper.

With the statement "gaps in our understanding of the complex interdisciplinary issue that is drought" we meant to refer to the second issue you mention, namely that there are parts of the complex interdisciplinary system that we do not understand. We did not mean to say that most hydrologists do not know that drought is complex. We see that this statement is open for multiple interpretations, especially after the examples we provided. We agree with your explanation that there also is a lack of ontology and semantics (and additionally a lack of two-way communication between stakeholders

and researchers), which results in a lack of understanding with the general public, policy makers and part of the impact communities. We have not touched upon this issue in our paper, but we certainly agree that it is good to mention. In a revised version of the paper we will rephrase the statement about the understanding of the drought issue.

We certainly support the second point you made about challenges for the current treatment of drought, i.e. the necessary extension of the drought research into including human impacts on drought, human response to droughts, incl. the socio-economic and environmental impacts. This is in line with the main message of the paper. We will stress this even more in the revised document.

We will address your minor points in our review. Here we only want to comment on your second and third point regarding definitions:

- Human activities reducing drought (hazard or impacts) are indeed important. This can be encompassed in the definition of "human-modified drought", because that means 'a drought changed by human activities'. We mention on p.4 l.28 that "human-modified drought" is "a drought that is enhanced or alleviated as the result of anthropogenic processes". This is however not indicated in Fig. 3, which we will adapt to include this alleviating influence of human activities on drought by adding a fourth drought event.

- The normal conditions mentioned in this paragraph do not imply a long-term balance of fluxes. It is not realistic to assume long-term balance of fluxes for any catchment, especially not those with high human influence. With the sentence "the desired situation is out of balance with the normal situation" we mean to say that water demand is higher than water availability. We now realise that this sentence needs some rephrasing to avoid confusion.

---

## Referee Comment (RC1) · Anonymous Referee #1 · 26 Jul 2016

Dear Editor and Authors,

In this draft opinion paper, the authors defined the term "drought in Anthropocene" and discussed the direction of future research on the topic. This is an opinion paper, but it is also a review paper on this research field. It includes both the views of eminent researchers and massive volume of information on the latest achievements and limitations of drought research which is potentially useful for the readers of HESS.

This is my first time to review an opinion paper. Because it basically conveys the authors' opinion, I have no strong recommendation on it. Below, I would like to express my honest impression of this paper which may be potentially useful for the authors for further revision.

I read through this draft paper for three times, but I still cannot fully grasp some of the authors' key meaning. Perhaps this is due to unclearness in the definition of "drought in Anthropocene" (Section 2) and the authors' attitude to include everything of drought (Sections 3 and 4). The latter point is a virtue of the authors, but some clear focuses would enhance readability.

After all, I couldn't fully understand the definition of the term "drought in Anthropocene" by the authors. First, although the authors used three pages to explain this term (Section 2), I couldn't find its conclusive definition. I understood that they newly proposed the concept of "human-modified drought" (page 4 lines 22-30), but is this the "drought in Anthropocene"? My confusion is originated from my belief that drought is a perception of humanity and society hence drought cannot be defined without humans. What does "drought in Anthropocene" exclusively mean? Next I couldn't clearly understand what "human modification of drought" means. The term "drought" indicates dry events that rarely occur (drier than normal, as the authors describe in Section 2.2). I found it meaningful to attribute the rareness to "human (i.e. the events that would not be happened if no humanity were there)" and "climate (i.e. the events that would be happened regardless the existence of humanity)". On the other hand, "human modification" basically occurs chronically. Why did the authors link this chronic state with probabilistically rare events? I guess this must be explained in text already, but finally I couldn't figure it out what are the authors' logic.

In Sections 3-4, the authors made great efforts to cover the diversity of drought issues from the scales of catchment to global, from the events of several days to several years, from moderate to extreme, from place to place. I agree with the authors that human interventions are seen every time and everywhere in the present world, but I believe their magnitude significantly differ by scale, period, events, and places. I guess humans play critical and exclusive roles in drought formation and consequence in a limited number of cases. It could be more informative and readable if the authors focus on the drought that humans' influences are obvious.

Finally, I largely agree with the comments by Prof. Marc Bierkens. Although the latest hydrological analyses or models do not incorporate many of human aspects, it does not necessarily mean we do not understand the mechanism, consequences, and countermeasures of drought. Indeed, local water managers and citizens are often aware of them well. This paper would become further important if it is narrowed down what are the unknowns and what should be investigated by researchers.

Specific comments

Page 4 line 4 "Drought as a lack of water": The subtitle doesn't well reflect the key contents of subsection. I got similar impression for other subsections. Please revisit all the subtitles in text.

Page 4 line 32 "Furthermore, the terms we propose are not new": Then what were newly added?

Page 5 line 16 "average water levels": This is unclear and confusing. Do you mean mean-annual water availability here?

Page 5 line 25 "Drought is often confused with water scarcity and water shortage": It would be highly helpful to add a table of conclusive definitions for drought in Anthropocene, water stress, and water scarcity, and relevant terms.

Page 5 line 31 "complemented": "Coincided"?

Page 5 line 31 "with short-term drought": Do you mean climate-induced drought here?

Page 6 line 9 "We have to point out that the definitions of drought and its impacts used here deviate from the definitions used in other scientific disciplines...": I've read this paragraph again and again, but I couldn't figure out the similarity and difference clearly. Again, it would be useful for readers if you add a table of definitions on drought and its impacts by disciplines.

Page 9 line 21 "resulting outflows depend on operational rules": A relevant study is

reported also in Mateo et al. (2014).

Page 10 line 4 "Impacts of drought in the Anthropocene": In my impression the section describes general impacts of drought, not specific to Anthropocene. Since drought is a perception of humanity, drought cannot be defined without humans. What is the key difference from the traditional definition of drought?

Page 12 line 18 "One novel type of qualitative data is the use of drought narratives": Is the data qualitative or quantitative?

Page 12 line 25 "classic water management models" What are the classic models? What differentiates from classic and another?

Page 13 line 9 "3.3" reads "3.5".

Page 14 line 2 "continuous increases in abstraction, and step-changes in storage by dam building": For instance, the works by Wisser et al. (2010) and Pokhrel et al. (2012) reported quantitatively simulations.

References

Mateo, C. M., Hanasaki, N., Komori, D., Tanaka, K., Kiguchi, M., Champathong, A., Sukhapunnaphan, T., Yamazaki, D., and Oki, T.: Assessing the impacts of reservoir operation to floodplain inundation by combining hydrological, reservoir management, and hydrodynamic models, Water Resour. Res., 50, 7245-7266, 10.1002/2013wr014845, 2014.

Wisser, D., Fekete, B. M., Vörösmarty, C. J., and Schumann, A. H.: Reconstructing 20th century global hydrography: a contribution to the Global Terrestrial Network- Hydrology (GTN-H), Hydrol. Earth Syst. Sci., 14, 1-24, 10.5194/hess-14-1-2010, 2010.

Pokhrel, Y. N., Hanasaki, N., Yeh, P. J. F., Yamada, T. J., Kanae, S., and Oki, T.: Model estimates of sea-level change due to anthropogenic impacts on terrestrial water storage, Nature Geosci., 5, 389-392, 10.1038/ngeo1476, 2012.

---

## Referee Comment (RC2) · Anonymous Referee #2 · 3 Aug 2016

This review paper discusses drought in the Anthropocene and identifies a number of important research gaps along with tools and techniques to start tackling these research questions. I really enjoyed reading the paper and agree with many of the points that were raised. I think the other two reviewers have highlighted several important points to be addressed so I just have a few comments to add.

Specific Comments

Section 3.2. P9 L 30. I agree that one of the key limiting factors is availability of data and metadata on human modifiers. However, even if this data is available there are still many questions on how to incorporate these processes into hydrological models. For example, how do we incorporate groundwater abstractions into current lumped or

semi-distributed hydrological models (often used for prediction of drought)? While the acquisition of data on human modifiers is important, this needs to be in conjunction with developing models (both catchment, regional and global scale models) that can make use of this data.

Section 3.3. This section was not as clear as the others in how we can move research forward in this area (e.g. how could we use impact information better? P 11, L 17). I would add a sentence or two to better clarify exactly what is needed to address the research gaps you highlighted at the beginning of section 3.3.

Section 3.5. I agree with Marc that you should use a different word to 'norm'.

Figure 4. I found Figure 4 and Table 1 quite confusing and not overly useful in helping me to understand the text – can you merge the two together to make a figure that is more easily understandable?

Figure 6a. Should there be a y-axis on this plot?

---

## Author Comment (AC2) · 4 Aug 2016

Thanks for your positive evaluation of our paper. In this reply we will shortly discuss your comments. A more detailed reply will follow with the revised version of our paper.

You mention that you have some issues understanding Section 2 "Defining drought in the Anthropocene". I think your confusion might have arisen from a misunderstanding of the title of this section. We say on p.2 l.1-2 that in this section "we revisit drought definitions and make suggestions for robust use in the Anthropocene". So we are not defining a new term called "drought in the Anthropocene", but we are evaluating how we should use existing drought definitions in the Anthropocene. Would your issue be solved if we rephrase the title of Section 2 as "Use of drought definitions in the

Anthropocene"?

With the term "human-modified drought" we indicate drought events that are caused by a combination of climate and human processes or drought events of which the severity has been altered substantially by human activities. These human processes / activities can be short-lived ("rare events" such as emergency relief groundwater abstraction) or more chronic (land use change, dam building). The latter are regarded in this paper as human activities changing the propagation of drought (see Figure 2), so they are not causing the specific event, but change its characteristics. We think it is important to make this explicit by using the term "human-modified drought", so that in the drought analysis the characteristics of the drought are attributed to the correct processes, which is crucial in drought management (see our examples in Box 1).

You also suggest to limit the information in Sections 3-4 by focussing on cases where drought is significantly influenced by human activities. We agree that the magnitude of human influence on drought is probably very variable, but since it has rarely been quantified an objective selection cannot easily be made. With this opinion / review article we hope to encourage colleagues to do that quantification of the relation between people and drought, so that in the future we will be able to make the selection you propose of which areas / processes are more important.

Thanks for your view on the comments of Prof Marc Bierkens. We agree that local water managers and citizens often have a lot of knowledge on local drought processes and their relation with human activities. In our paper we therefore suggest to make more use of this knowledge, e.g. p.9 l.5-10 where we advise that "More qualitative and local scale information on the human influences in a catchment can be gathered by a range of methods, including . . ." and p.12 l.18-24 where we state that "Qualitative data (such as drought narratives) are essential in our quest for increased understanding" of feedbacks between drought and society.

We will address your specific comments in our review. Here we only comment on your

point regarding section titles and definitions:

- We will revisit the section and subsection titles now we are aware that they can create confusion. We agree that our section on "Impacts of drought in the Anthropocene" is not specific for the Anthropocene. Quantifying the relation between drought characteristics and drought impacts is however a very relevant open research question in the Anthropocene. We suggest to remove the "in the Anthropocene" from the subsection titles, so that the structure of this section becomes: "Drought drivers", "Modifications of drought", "Drought impacts", "Feedbacks between humans and drought" and "Changing norms".

- You suggest to add a table with definitions for drought, water scarcity and other relevant terms. Such a table is already included (Table 1). We will include more references to Table 1 in the text, where needed. We will also rewrite the paragraph on definitions in other disciplines to clarify the difference.

- Thanks for the suggested references.
* * *

---

## Author Comment (AC3) · 4 Aug 2016

Thanks for your positive evaluation of our paper. Here we will shortly respond your comments. In the revised version of our paper we will make the suggested changes.

- We agree that it is still unclear how to incorporate human processes correctly into current lumped or semi-distributed hydrological models. In Section 3.2, we have only given examples of recent developments in large-scale models, but we agree that it is needed to also briefly mention other types of hydrological models, especially those that are used as prediction tools in local catchments. We do mention water management models in Section 3.4 (p.12, l.25-31), but in a revised version of the paper we will add some sentences on incorporating human processes into lumped or semi-distributed

hydrological models to Section 3.2 as well. Thanks for the suggestion.

- We will revisit Section 3.3. We will rephrase the research questions and add more information on how to best solve these with future research.

- We agree too and will make the suggested change in the revised version of the manuscript.

- We will have a look again at Figure 4 and Table 1. In the preparation of the manuscript these have already provided significant challenges and multiple versions of both have been discussed between the co-authors. Nevertheless, we will attempt to merge Figure 4 and Table 1 and make them more useful.

- We will redo figure 6a.

---

## Author Response (AR1)

We want to thank the editor, two reviewers and Marc Bierkens for their positive evaluation of our paper. In this response we copy the reviewer comments, repeat our previous replies and give a point-by-point overview of the changes we made in the revised version of our paper. Our comments and changes are in *italics* and preceded by ">>". The page and line numbers we mention refer to the attached track changes version of the manuscript, unless stated otherwise. The track-changes version of the manuscript is included at the end of this rebuttal.

**Marc Bierkens**

I thoroughly enjoyed reading this paper and subscribe to most of its views. Indeed, it is time that we start including the human dimension into drought analyses and prediction. The paper sets up a comprehensive framework for defining the various types of drought in the Anthropocene and identifies a number of research gaps. There is one major point I would like to make about the framing of the paper. It is claimed on page 3, line 14 that there are "gaps in our understanding of the complex interdisciplinary issue that is drought". This seems to suggest that we as a community of hydrologist are unaware of the fact that drought is a many-headed dragon. I beg to differ here. I am not sure it is lack of understanding per se. It depends on who the "we" are in "our understanding". Surely, hydrologists that are used to simulating or studying the hydrological system holistically, understand how the interactions between the hydrological states and fluxes and human intervention on these work conjunctively. It seems to me that it is more the general public and the policy makers that don't know or understand, partly because we as scientists lack a proper description of the interacting parts (ontology) or use the incorrect terms (semantics). So the question is not so much as "do we (as hydrologists) lack understanding?" but rather "do we as scientists lack the correct ontology and terminology for drought in the Anthropocene and therefore fail to convey understanding with the general public and the policy makers?".

This being said, this does not mean that all issues are resolved. Indeed, if we acknowledge that the study of drought also encompasses the human dimension and we also extend drought impacts from mere indices or water volumes to socio-economic and ecological impacts and if we want to understand how human water demand reacts to extreme events or a changing normal in the future, we need additional knowledge, i.e. understanding of additional system parts. In conclusion: I would set the stage a bit differently: there are two problems with current treatment of drought:

1) Lacking the correct ontology and associated semantics to describe what makes a drought, which results in lack of understanding of (the causes of) drought of public and policy makers.

2) Necessary extension of the drought research into including human impacts on drought, human response to droughts and the socio-economic and ecological effects on drought that leads to a number (you name 5) areas with lack of understanding of the (inner workings of) parts of the system.

*>> Thanks for sharing these thoughts. With the statement "gaps in our understanding of the complex interdisciplinary issue that is drought" we meant to refer to the second issue you mention, namely that there are parts of the complex interdisciplinary system that we do not understand. We did not mean to say that most hydrologists do not know that drought is complex. We see that this statement*

*is open for multiple interpretations, especially after the examples we provided. We agree with your explanation that there also is a lack of ontology and semantics (and additionally a lack of two-way communication between stakeholders and researchers), which results in a lack of understanding with the general public, policy makers and part of the impact communities. We have not touched upon this issue in our paper, but we certainly agree that it is good to mention. In the revised version of the paper we rephrased the statement about the understanding of the drought issue as follows (p.3 l.19-24): "These examples point out a number of issues (see Box 1). Firstly, recent (drought) research is not always picked up by water managers and policy makers. There exists a lack of two-way communication between stakeholders and researchers, with proper ontology and semantics. Secondly, drought research itself has some important gaps related to the interplay between drought and humans, which prevent us from completely understanding the complex interdisciplinary issue that is drought. Thirdly, these examples also highlight the unsuitability of current methods and data to address these gaps."*

*>> We also added a sentence to the conclusion of the paper, which reads: "For improved drought management in the Anthropocene, a better two-way communication between scientists, stakeholders, policy makers and general public is needed. There are often social, psychological and organisational barriers that prevent optimal use of scientific understanding in decision making. They not our primary focus here, but clearly they can play an important role." (p.16 l.27-30). Additionally, also in response to the comments of reviewer 1, we changed Table 1 so that it now encompasses a list of definitions of drought and related terms, i.e. addressing your point of the semantics.*

*>> We certainly support the second point you made about challenges for the current treatment of drought, i.e. the necessary extension of the drought research into including human impacts on drought, human response to droughts, incl. the socio-economic and environmental impacts. This is in line with the main message of the paper. We stressed this even more in the revised document (p.3 l.21-22 and p.4 l.1).*

Some minor points to make:

*>> We addressed all your minor points in our review.*

- Page 3, line 12: "Hydrological drought?" *>> Yes. We changed this.*

- Page 4, lines 26-28: Great definition. But what if human activities lead to reduced drought? Think about the impacts of dams on low flows (your Figure 5) or return flows from deep (industrial) groundwater abstractions that also increase discharge in low flow periods (see de Graaf et al., Advances in Water Resources 2013).
  *>> Human activities reducing drought (hazard or impacts) are indeed important. This can be encompassed in the definition of "human-modified drought", because that means 'a drought changed by human activities'. We mention on p.4 l.28 that "human-modified drought" is "a drought that is enhanced or alleviated as the result of anthropogenic processes". This is however not indicated in Fig. 3, which we adapted to include this alleviating influence of human activities on drought by adding a fourth drought event. Based on the comments of reviewer 1 we also changed Table 1, so that it now restates the definition of human-modified*

*drought. We also added the reference of de Graaf et al. 2014 (p.9 l.27 & p.10 l.25) and added some sentences to Section 2.2 stressing also the positive effect of humans on drought (p.5 l.26-27 & p.6 l.2-4).*

- Page 5, line 30: Here there is a problem with your definitions. Before you defined "normal" as long term average under human and natural conditions. You should then add to the conditions that define a normal that there is long-term balance of fluxes.
  *>> The normal conditions mentioned in this paragraph do not imply a long-term balance of fluxes. It is not realistic to assume long-term balance of fluxes for any catchment, especially not those with high human influence. With the sentence "the desired situation is out of balance with the normal situation" we mean to say that water demand is higher than water availability. We now realise that this sentence needs some rephrasing to avoid confusion. We added an explanation to the sentence so that it now reads: "desired situation is out of balance with the normal situation, i.e. average water demand is higher than average water availability, …" (p.6 l.19-20).*

- Page 12, line 24: remove the reference or find another. It is of no use to refer to an in prep paper.
  *>> The paper is now in review. We added the complete reference to the reference list.*

- Page 12, lines 28-31 "Many water management. . …scenario (Watts et al., 2012)". Actually that is also the case for many of the global hydrology and water resources models such as PCR-GLOBWB where water abstractions are a function of water availability".
  *>> We agree that some (global) hydrological models are also able to simulate some form of water allocation, although often not as detailed as water resources models. We changed the sentence to include a reference to other hydrological models: "Like some global and lumped hydrological models mentioned before, many water management models, however, are capable of simulating…" (p.14 l.4-5).*

- Page 13: the section title "Changing norm in the Anthropocene". Given the terminology you use, it is not better to use "changing normal" (see also your Figure 4 and the use of the term "normal" in meteorology)? A norm has also a normative connotation.
  *>> We changed "norm" to "normal", "normal situation" or "reference situation" throughout the whole paper.*

**Reviewer #1:**

Dear Editor and Authors, In this draft opinion paper, the authors defined the term "drought in Anthropocene" and discussed the direction of future research on the topic. This is an opinion paper, but it is also a review paper on this research field. It includes both the views of eminent researchers and massive volume of information on the latest achievements and limitations of drought research which is potentially useful for the readers of HESS. This is my first time to review an opinion paper. Because it basically conveys the authors' opinion, I have no strong recommendation on it. Below, I

would like to express my honest impression of this paper which may be potentially useful for the authors for further revision.

I read through this draft paper for three times, but I still cannot fully grasp some of the authors' key meaning. Perhaps this is due to unclearness in the definition of "drought in Anthropocene" (Section 2) and the authors' attitude to include everything of drought (Sections 3 and 4). The latter point is a virtue of the authors, but some clear focuses would enhance readability. After all, I couldn't fully understand the definition of the term "drought in Anthropocene" by the authors. First, although the authors used three pages to explain this term (Section 2), I couldn't find its conclusive definition. I understood that they newly proposed the concept of "human-modified drought" (page 4 lines 22-30), but is this the "drought in Anthropocene"? My confusion is originated from my belief that drought is a perception of humanity and society hence drought cannot be defined without humans. What does "drought in Anthropocene" exclusively mean? Next I couldn't clearly understand what "human modification of drought" means. The term "drought" indicates dry events that rarely occur (drier than normal, as the authors describe in Section 2.2). I found it meaningful to attribute the rareness to "human (i.e. the events that would not be happened if no humanity were there)" and "climate (i.e. the events that would be happened regardless the existence of humanity)". On the other hand, "human modification" basically occurs chronically. Why did the authors link this chronic state with probabilistically rare events? I guess this must be explained in text already, but finally I couldn't figure it out what are the authors' logic.

*>> We think your confusion might have arisen from a misunderstanding of the title of this section. We say on p.2 l.1-2 (referring to previous version of paper) that in this section "we revisit drought definitions and make suggestions for robust use in the Anthropocene". So we are not defining a new term called "drought in the Anthropocene", but we are evaluating how we should use existing drought definitions in the Anthropocene. We now rephrased the title of Section 2 as "Drought definitions in the Anthropocene" (p.4 l.3).*

*>> With the term "human-modified drought" we indicate drought events that are caused by a combination of climate and human processes or drought events of which the severity has been altered substantially by human activities. These human processes / activities can be short-lived ("rare events" such as emergency relief groundwater abstraction) or more chronic (land use change, dam building). The latter are regarded in this paper as human activities changing the propagation of drought (see Figure 2), so they are not causing the specific event, but change its characteristics. We think it is important to make this explicit by using the term "human-modified drought", so that in the drought analysis the characteristics of the drought are attributed to the correct processes, which is crucial in drought management (see our examples in Box 1). This is now better clarified in Table 1.*

*>> In Section 2.2 we rephrased the option to evaluate the normal or reference situation in the Anthropocene to clarify "human modified drought" in relation to chronic human influence (p.5 l.28 – p.6 l.4). We changed Table 1, so that it now explains the definitions of drought and some related terms. In response to the comments of Marc Bierkens we also adapted Fig. 3 to include the alleviating influence of human activities on drought by adding a fourth drought event to the figure.*

In Sections 3-4, the authors made great efforts to cover the diversity of drought issues from the scales of catchment to global, from the events of several days to several years, from moderate to extreme, from place to place. I agree with the authors that human interventions are seen every time and everywhere in the present world, but I believe their magnitude significantly differ by scale, period, events, and places. I guess humans play critical and exclusive roles in drought formation and consequence in a limited number of cases. It could be more informative and readable if the authors focus on the drought that humans' influences are obvious.

*>> We agree that the magnitude of human influence on drought is probably very variable, but since it has rarely been quantified an objective selection cannot easily be made. With this opinion / review article we hope to encourage colleagues to do that quantification of the relation between people and drought, so that in the future we will be able to make the selection you propose of which areas / processes are more important.*

Finally, I largely agree with the comments by Prof. Marc Bierkens. Although the latest hydrological analyses or models do not incorporate many of human aspects, it does not necessarily mean we do not understand the mechanism, consequences, and countermeasures of drought. Indeed, local water managers and citizens are often aware of them well. This paper would become further important if it is narrowed down what are the unknowns and what should be investigated by researchers.

*>> Thanks for your view on the comments of Prof Marc Bierkens. We agree that local water managers and citizens often have a lot of knowledge on local drought processes and their relation with human activities. In our paper we therefore suggest to make more use of this knowledge, e.g. p.9 l.5-10 (referring to previous version of paper) where we advise that "More qualitative and local scale information on the human influences in a catchment can be gathered by a range of methods, including …" and p.12 l.18-24 (referring to previous version of paper) where we state that "Qualitative data (such as drought narratives) are essential in our quest for increased understanding" of feedbacks between drought and society. We added the reference of Daniels and Endfield (2009) to our statement about narratives (p.13 l.28-29).*

We addressed your specific comments in our review:

- Page 4 line 4 "Drought as a lack of water": The subtitle doesn't well reflect the key contents of subsection. I got similar impression for other subsections. Please revisit all the subtitles in text.
  *>> We revisited the section and subsection titles now we are aware that they can create confusion. We removed the "in the Anthropocene" from the subsection titles, so that the structure of this section becomes: "Drought drivers", "Modifications of drought", "Drought impacts", "Feedbacks between humans and drought" and "Changing normal".*

- Page 4 line 32 "Furthermore, the terms we propose are not new": Then what were newly added?

*>> As explained above, no new terms were added.*

- Page 5 line 16 "average water levels": This is unclear and confusing. Do you mean mean-annual water availability here?
*>> Not exactly. We want to stress that we are analysing measured (or modelled) water levels and fluxes (which we added now). Also, the average does not need to be annual, but is calculated on the time scale used for the analysis (e.g. daily / monthly). What we did forget to mention is that we refer to long-term averages. So, to avoid confusion, we rephrased this sentence to "long-term average water levels or fluxes" (p.6 l.5).*

- Page 5 line 25 "Drought is often confused with water scarcity and water shortage": It would be highly helpful to add a table of conclusive definitions for drought in Anthropocene, water stress, and water scarcity, and relevant terms.
*>> We adapted Table 1. It now includes definitions for drought, water scarcity and other relevant terms. We also included more references to Table 1 in Section 2.2 and 2.3.*

- Page 5 line 31 "complemented": "Coincided"?
*>> Yes, thank you. We changed this sentence to "it coincides with" (p.6 l.21).*

- Page 5 line 31 "with short-term drought": Do you mean climate-induced drought here?
*>> No, not necessarily. We mean to illustrate the difference between long-term and short-term influences. In this case it can refer to the short-term effects of climate-induced or human-induced drought.*

- Page 6 line 9 "We have to point out that the definitions of drought and its impacts used here deviate from the definitions used in other scientific disciplines." I've read this paragraph again and again, but I couldn't figure out the similarity and difference clearly. Again, it would be useful for readers if you add a table of definitions on drought and its impacts by disciplines.
*>> It is all a matter of reference. For the climate community, DROUGHT is an IMPACT of extreme climate, whereas for the hydrological community DROUGHT is a state of the physical system that CAUSES IMPACTS on society and ecosystems. We rewrote the paragraph on definitions in other disciplines to clarify the difference (p.6 l.32 – p.7 l.4). We don't think this paragraph needs a table to show the difference in definitions and terms used by different communities. That would be outside the scope of this paper. We only wanted to highlight that there are differences in phrasing between communities and gave the example of the climate community because that community dominates the drought concourse.*

- Page 9 line 21 "resulting outflows depend on operational rules": A relevant study is reported also in Mateo et al. (2014).
*>> Thanks, we added the reference (p.10 l.17)*

- Page 10 line 4 "Impacts of drought in the Anthropocene": In my impression the section describes general impacts of drought, not specific to Anthropocene. Since drought is a

perception of humanity, drought cannot be defined without humans. What is the key difference from the traditional definition of drought?

*>> We agree that our section on "Impacts of drought in the Anthropocene" is not specific for the Anthropocene. Quantifying the relation between drought characteristics and drought impacts is however a very relevant open research question in the Anthropocene. As mentioned above we removed the words "in the Anthropocene" from all the subsection titles.*

- Page 12 line 18 "One novel type of qualitative data is the use of drought narratives": Is the data qualitative or quantitative?

  *>> The drought narratives data is qualitative. We added a description of what we mean with drought narratives and a reference to a paper in which they are used (p.13 l.28-29).*

- Page 12 line 25 "classic water management models" What are the classic models? What differentiates from classic and another?

  *>> We mean water management models that only model the allocation of available water to water users, without simulating in detail hydrological processes. We removed "classic" because the next sentence explains what we mean (p.14 l.1).*

- Page 13 line 9 "3.3" reads "3.5".

  *>> Thanks. We changed 3.3 to 3.5.*

- Page 14 line 2 "continuous increases in abstraction, and step-changes in storage by dam building": For instance, the works by Wisser et al. (2010) and Pokhrel et al. (2012) reported quantitatively simulations.

  *>> Thanks for the suggested references. We added them to the manuscript (p.15 l.12).*

**Reviewer #2:**

This review paper discusses drought in the Anthropocene and identifies a number of important research gaps along with tools and techniques to start tackling these research questions. I really enjoyed reading the paper and agree with many of the points that were raised. I think the other two reviewers have highlighted several important points to be addressed so I just have a few comments to add.

Specific Comments

- Section 3.2. P9 L 30. I agree that one of the key limiting factors is availability of data and metadata on human modifiers. However, even if this data is available there are still many questions on how to incorporate these processes into hydrological models. For example, how do we incorporate groundwater abstractions into current lumped or semi-distributed hydrological models (often used for prediction of drought)? While the acquisition of data on human modifiers is important, this needs to be in conjunction with developing models (both catchment, regional and global scale models) that can make use of this data.

*>> We agree that it is still unclear how to incorporate dynamic and responsive human processes correctly into current lumped or semi-distributed hydrological models. In Section 3.2, we have only given examples of recent developments in large-scale models, but we agree that it is needed to also briefly mention other types of hydrological models, especially those that are used as prediction tools in local catchments. We do mention water management models in Section 3.4 (p.12, l.25-31), but in the revised version of the paper we added some sentences on incorporating human processes into lumped or semi-distributed hydrological models to Section 3.2 as well: "Most predictions on local scale, however, are done with lumped or semi-distributed hydrological models. It is often not straightforward to incorporate dynamic human processes into these models and more work is needed to adapt these lumped hydrological models for use in the Anthropocene. An example of a new lumped hydrological model that incorporates man-made extraction and supply of water to both surface and subsurface water is WALRUS by Brauer et al. (2014). Current physically-based models are better fitted to simulate human responses to drought, e.g. SIMGRO (Querner et al., 2008; Van Lanen et al., 2004)." (p.10 l.29-34) Thanks for the suggestion.*

- Section 3.3. This section was not as clear as the others in how we can move research forward in this area (e.g. how could we use impact information better? P 11, L 17). I would add a sentence or two to better clarify exactly what is needed to address the research gaps you highlighted at the beginning of section 3.3.

  *>> We revisited Section 3.3. The text was slightly reorganised to better highlight the two aspects of 'data' on impacts and the 'link function' between drought indices and impacts as well as the consideration of vulnerability, and amended by more suggestions for future work (p.11 l.21 – p.12 l.7).*

- Section 3.5. I agree with Marc that you should use a different word to 'norm'.

  *>> We agree too and changed "norm" to "normal", "normal situation" or "reference situation" throughout the whole paper.*

- Figure 4. I found Figure 4 and Table 1 quite confusing and not overly useful in helping me to understand the text – can you merge the two together to make a figure that is more easily understandable?

  *>> Also based on the comments of reviewer 1, we adapted Table 1 and Figure 4. Table 1 now includes the definition of terms related to drought, aridity, water scarcity and overexploitation. Figure 4 visualises these concepts in different climate regions and under natural and human-influenced conditions. We hope that this new Table and Figure are more useful in helping to understand the text.*

- Figure 6a. Should there be a y-axis on this plot?

  *>> We redid figure 6 for clarity. Figure 6a does not have a y-axis because it is a conceptual figure showing threshold behaviour and non-linearity of the climate-yield relationship.*

[revised manuscript text omitted]